# Effect of Transcutaneous Auricular Vagus Nerve Stimulation in Chronic Low Back Pain: A Pilot Study

**DOI:** 10.3390/jcm13247601

**Published:** 2024-12-13

**Authors:** Isabelle Tavares-Figueiredo, Yves-Marie Pers, Claire Duflos, Fanchon Herman, Benjamin Sztajnzalc, Hugo Lecoq, Isabelle Laffont, Arnaud F. Dupeyron, Alexis F. Homs

**Affiliations:** 1Department of Physical Medicine and Rehabilitation, CHU Montpellier, University of Montpellier, 34295 Montpellier, France; i-tavaresfigueiredo@chu-montpellier.fr (I.T.-F.); benjamin.sztajnzalc@gmail.com (B.S.); hugo-lecoq@hotmail.fr (H.L.); isabelle.laffont@umontpellier.fr (I.L.); 2Centre d’Investigation Clinique, CHU Montpellier Montpellier, Inserm, CIC 1411, 34295 Montpellier, France; 3IRMB, University of Montpellier, INSERM, 34295 Montpellier, France; ym-pers@chu-montpellier.fr; 4Clinical Immunology and Osteoarticular Diseases Therapeutic Unit, Lapeyronie University Hospital, CHU Montpellier, 34295 Montpellier, France; 5Clinical Research and Epidemiology Unit, CHU Montpellier, University of Montpellier, 34295 Montpellier, France; c-duflos@chu-montpellier.fr (C.D.); fanchon.herman@chu-montpellier.fr (F.H.); 6EuroMov Digital Health in Motion, University of Montpellier, IMT Mines Ales, 34090 Montpellier, France; arnaud.dupeyron@umontpellier.fr; 7Department of Physical Medicine and Rehabilitation, CHU Nimes, University of Montpellier, 30900 Nimes, France

**Keywords:** vagus nerve stimulation, chronic low back pain, pain, disability, non-pharmacological therapy

## Abstract

**Background/Objectives**: Chronic low back pain (CLBP) is a common condition with limited long-term treatment options. Vagus nerve stimulation (VNS) has shown potential for pain improvement, but its use in CLBP remains underexplored. Our aim was to evaluate the efficacy, feasibility and tolerability of transcutaneous auricular vagus nerve stimulation (taVNS) in reducing pain and improving functional outcomes in CLBP patients. **Methods**: Thirty adults with CLBP (VAS ≥ 40/100) participated in this open-label pilot study (NCT05639270). Patients were treated with a taVNS device on the left ear for 30 min daily over a period of 3 months. The primary outcome was a reduction in pain intensity (VAS) at 1 month. Secondary outcomes included pain intensity at 3 months, disability (Oswestry Disability Index, ODI), quality of life (EQ-5D-5L), catastrophizing and psychological distress. In addition, compliance and adverse events were monitored. **Results**: After 1 month, 27 patients were evaluated. VAS scores decreased significantly by 16.1 (SD = 17.9) mm (*p* < 0.001) and by 22.5 (25) mm (*p* < 0.001) after 3 months (24 patients were analyzed). Functional disability improved with an average reduction in ODI of 11.9 (11.1) points (*p* < 0.001) after 3 months. Other patient-reported outcomes also improved significantly over the 3-month period. Overall, 51.9% of the patients achieved clinically meaningful pain reduction (≥20 mm), and no serious adverse events were reported. Treatment adherence was good, with half of the patients achieving 80% adherence. **Conclusions**: This pilot study suggests that taVNS is a feasible, safe and potentially effective treatment for CLBP that warrants further investigation in a randomized controlled trial compared to sham stimulation.

## 1. Introduction

Chronic low back pain (CLBP) is one of the most common and disabling conditions affecting a significant proportion of the world’s population [1]. It is a major cause of functional impairment, representing 7.7% of all years lived with disability, the greatest contribution to the world’s burden of disability [2]. Although a variety of treatment options are available, including medication, physiotherapy and surgery, many patients experience only limited relief [3]. The ongoing challenge of effectively treating CLBP and concerns about the long-term use of pharmacologic treatments—such as opioid dependence and side effects—highlight the urgent need for therapeutic innovation [4]. In particular, there is growing interest in non-pharmacologic approaches that can provide sustained, long-term relief without the risks associated with traditional drug therapies [5].

Neuromodulation techniques have emerged as potential therapeutic options for the treatment of chronic pain, offering a novel approach that targets the nervous system to modulate pain pathways [6]. Among these techniques, vagus nerve stimulation (VNS) has gained attention due to its ability to affect both the peripheral and central nervous systems [7]. VNS primarily activates parasympathetic pathways, exerts anti-inflammatory effects and modulates pain perception through interactions with the brainstem [8]. By reducing the release of pro-inflammatory cytokines and promoting autonomic balance, VNS may attenuate the neuroinflammatory processes thought to contribute to chronic pain conditions [9].

Transcutaneous auricular vagus nerve stimulation (taVNS), traditionally used as an invasive therapy for conditions such as epilepsy and treatment-resistant depression, has become a non-invasive alternative thanks to recent advances in medical devices [10]. This technique stimulates the auricular branch of the vagus nerve, which is accessible on the surface of the ear, making it a safer and more accessible option [11]. In transcutaneous auricular vagus nerve stimulation (taVNS), an electrode is placed on the cymba concha of the left ear [12]. This region is targeted for stimulation as it is innervated by the ascending auricular branch of the vagus nerve, and stimulation of the left side is preferred as it has a lesser effect on the sinoatrial node compared to the right branch [13]. Functional MRI studies have shown that stimulation of the cymba concha activates the first central relay of the vagus nerve, the nucleus tractus solitarius [10]. Previous studies have shown promising results of taVNS in various chronic pain conditions such as fibromyalgia, migraine and hand osteoarthritis [14,15,16,17], suggesting that it may also have therapeutic benefits in the treatment of CLBP.

To our knowledge, there are few studies looking at the effects of taVNS in CLBP and its role in this patient group has not been adequately explored. The only existing study is an unblinded trial in which taVNS was tested as an adjunct to exercise therapy in a small group of 11 patients over a period of only two weeks, with a very large effect size of 2.2, with no mention of the tolerability and feasibility of taVNS in the CLBP population [18].

We therefore decided to conduct a proof-of-concept open-label pilot study on a larger group of patients. Our main objective was to evaluate the effects of taVNS on pain intensity. We also wanted to measure the evolution of patient-reported functional CLBP outcomes and evaluate the feasibility and tolerability of taVNS in this patient group.

## 2. Materials and Methods

### 2.1. Study Design and Participants

The VALOM study (VAgus nerve stimulation for LOw back pain Management) was a prospective, interventional, open, uncontrolled study (NCT05639270).

Eligible participants were adults aged 18 to 70 years who had suffered from non-specific CLBP for more than three months, whose mean visual analog scale (VAS) pain intensity was at least 40 on a numeric scale of 0 to 100, who did not respond to well-performed physiotherapy and analgesics and for whom no change in treatment was planned in the month following inclusion.

Exclusion criteria were an ear canal incompatible with the stimulation device, the use of other electrical devices (e.g., pacemaker or transcutaneous electrical neurostimulation), a history of vagotomy, cardiac arrhythmia, the presence of a cochlear implant on the stimulation side, an existing or planned pregnancy during the study period, breastfeeding, participation in another biomedical research project, the presence of a legal guardian and low back pain caused by specific diseases (such as ankylosing spondylitis, spondylodiscitis or cancer). The procedures used were in accordance with the ethical standards of the competent committee for human experimentation (Comite de Protection des Personnes number 2022-AO1557-36) and the Declaration of Helsinki. All patients gave their written informed consent.

### 2.2. Procedures

The patients were recruited as part of routine examinations in the Department of Physical Medicine and Rehabilitation at Montpellier University Hospital and via an advertisement in a local newspaper. They were verbally informed of the conditions for participation in the study and an information letter summarizing the inclusion and exclusion criteria and the conduct of the study was sent by email. After verbal consent and review of the eligibility criteria, they were enrolled in the study. Upon admission, an electrocardiogram was performed to rule out cardiac arrhythmias. The patients were then administered the taVNS device (VAGUSTIM device, Schwa Medico, Rouffach, France, CE0197-2019/04/24) together with an ear stimulation electrode, a tube of conductive gel and earplugs of different sizes to adapt to the anatomical differences of the pinna. Figure 1 demonstrates the placement of the taVNS device, showing electrode positioning relative to the auricular anatomy. The taVNS device delivered a continuous current with a biphasic, asymmetric and balanced waveform and was set up for 30 min of daily stimulation at a frequency of 25 Hz and a pulse width of 50 µs. An initial switch-on session was conducted with the patient, with advice given to determine the ideal stimulation threshold in mA to achieve sensory stimulation without discomfort or dysesthesia [13]. The recommended stimulation intensity in mA was not fixed but consisted of stimulation that the patient perceived as not painful. Each patient was informed that in order to comply with the protocol, they had to perform a 30 min stimulation in the pinna of the left ear at any time of the day for a period of three months. Patients had the opportunity to contact the research team if any questions arose. Two follow-up visits (which lasted about 30 min in total) were scheduled after one and three months, including questionnaire completion and heart rate variability (HRV) measurements. Additionally, patients were called weekly during the first month to record the main outcome and the tolerability of the device.

### 2.3. Outcomes

The primary endpoint was change of the 0–100 mm VAS for low back pain (during the last 24 h) between baseline and one month. Secondary endpoints included VAS for low back pain at 3 months, the level of disability measured with the Oswestry Disability Index (ODI) [19], anxiety and depression measured with the Hospital Anxiety and Depression Scale (HADS) [20], quality of life measured with the EQ-5D-5L [21] and catastrophizing measured with the Pain Catastrophizing Scale (PCS) [22] at one and three months. A detailed description of the main characteristics of the questionnaires, including their validation in French and cutoff values, is provided in Appendix A. To determine whether a patient responder profile could be identified, the minimum clinically meaningful improvement was defined as 20 mm out of 100 for the absolute improvement of the VAS for low back pain [23].

To observe a possible trend towards a change in the patients’ sympathetic or parasympathetic profile, heart rate variability was measured with a chest strap device at baseline and after 1 and 3 months. HRV was measured at rest for five minutes at each visit [24,25]. Compliance was measured by recording the stimulation history (number of stimulations, their date and duration and the intensity thresholds in mA) on each device after completing the study. There is no precise definition of compliance when using a medical device. In conventional pharmacology, patients are considered compliant if they have taken 80% of their prescribed medication [26]. According to this definition, a compliant patient in our study should have completed at least 23 stimulations at the 1-month visit and 72 stimulations at the 3-month visit. Treatment credibility and satisfaction were assessed with a questionnaire consisting of six questions with answers ranging from 0 to 9, with a high total score indicating a high level of confidence in symptom improvement. Finally, in terms of treatment tolerance, patients were asked to report any adverse events (local or general) during follow-up.

### 2.4. Statistical Analysis

The quantitative variables are reported using mean and standard deviation and the qualitative variables are reported using frequency and percentage. The variation of VAS between baseline and 4 weeks was performed using a paired Student’s *t*-test based on the normal distribution (Shapiro–Wilk test). Longitudinal analysis was also performed using linear mixed model (LMM) that modelled the change in VAS over time. This model combined a fixed time effect and a random effect: a random intercept which takes into account the correlation between the different observations for the same patient.

Variations (between baseline and 1 month and between baseline and 3 months) of secondary endpoints were performed using either Student’s *t*-test for paired series or a Wilcoxon–Mann–Whitney test paired with the Shapiro–Wilk test. Analyses on independent samples (observer vs. non-observer and responder vs. non-responder) were performed with either Student’s *t*-test or a Wilcoxon–Mann–Whitney test based on the normal distribution (Shapiro–Wilk test) for quantitative variables. For qualitative variables, exact Fisher test was used. A sample size of 30 participants was chosen based on practical considerations and recommendations suggesting that 20 to 30 participants are sufficient to assess feasibility, estimate variability and detect medium-to-large effect sizes in pilot studies [27].

All tests were two-tailed tests, and a *p*-value < 0.05 indicates significance of the test. Statistical analyses were performed using SAS 9.4 (SAS Institute, Cary, NC, USA) software and R, version 4.3.1.

## 3. Results

Thirty patients were included between February and December 2023 in the university hospital of Montpellier. Their characteristics at inclusion are listed in Table 1. At baseline, 38% of the patients had severe CLBP according to the ODI scores, and 50% showed clinically significant catastrophizing according to the PCS scores. With regard to anxiety and depression, 60% and 20%, respectively, exceeded the clinically significant thresholds of the HADS scale.

Three patients were withdrawn from the study before the first month (not for safety reasons) and were therefore not included in the analysis of the results at 1 and 3 months. The first patient suffered from a fracture after a fall unrelated to vagal stimulation in the first month. The second patient suffered from a psoriasis flare with a skin lesion on the left ear outside the stimulation zone and stopped taVNS treatment before the one-month examination. The third patient suffered from an episode of debilitating sciatica, did not attend the one-month visit and underwent decompression surgery in the lumbar region.

At the one-month visit, another patient reported spikes in blood pressure associated with palpitations and discomfort after one week of using the device. At the patient’s request, we discontinued the study, and this patient was not included in the analysis of the results at 3 months. Finally, two patients participated in a standardized rehabilitation program during their participation in the study, after the 1-month visit; this introduced a bias in the analysis of secondary outcomes, particularly the VAS at 3 months, but did not affect the results of the primary outcome at 1 month. In total, 24 patients were analyzed at 3 months.

### 3.1. Efficacy

For the 27 patients analyzed at 1 month, the baseline VAS score for low back pain was 63 (SD = 13.9) mm. As shown in Table 2, taVNS significantly reduced the VAS by an average of 16.1 (17.9) mm after 1 month of use (*p* < 0.001). This effect increased in 24 patients until the 3-month visit, with a mean decrease of 22.5 (25) mm (*p* < 0.001). Figure 2 shows the evolution of the VAS for low back pain at each time point, including the weekly phone calls during the first month. We found a significant reduction in the VAS at each time point and a significant effect of time in an additional analysis with a linear mixed model (Appendix A).

Regarding clinical scores assessed by self-questionnaires, we also found a significant improvement in ODI score at 1 and 3 months (at 3 months: −11.9 (11.1), *p* < 0.001), quality of life (at 3 months: 14.2 (22.4), *p* = 0.01) and PCS total score (at 3 months: −8.6 (9.9), *p* < 0.001). The HADS total score improved significantly after 3 months (−2.7 (5), *p* = 0.02) (Table 2). The proportion of patients with severe disability on the ODI scores decreased to 18% and 15% after 1 and 3 months, respectively. A total of 25% and 21% of the patients had clinically significant catastrophizing at 1 and 3 months, respectively. For the HADS scores, 41% and 25% exceeded the thresholds for anxiety at 1 and 3 months and 11% and 8% exceeded the thresholds for depression at 1 and 3 months. Finally, no significant changes were observed in BMI, blood pressure, weekly duration of physical activity and tobacco consumption after either 1 or 3 months.

### 3.2. Responder Profile

With the aim of defining the profile of a patient likely to better respond to taVNS and potentially to guide prescribing in clinical practice, we set a threshold for a minimal clinically important difference (MCID) of 20 mm out of 100 for the VAS in low back pain at 1 month [23]. Fourteen of the twenty-seven patients (51.9%) achieved the MCID at 1 month. This group of patients with a better response showed a mean decrease of 30.3 (SD = 9.9) mm at 1 month and 36.9 (25.7) mm at 3 months compared to a mean decrease of 8.1 (13.7) mm at 3 months in the non-responder group. When comparing baseline characteristics between these two subgroups, the only significant differences were a lower total PCS score (21.6 (11.5) vs. 31.4 (9.5), *p* = 0.02) and less frequent lumbar steroid injections (28.6% vs. 76.9%, *p* = 0.02) in the responder group (Appendix A). History or current analgesics consumption, duration of CLBP, baseline pain VAS, demographic variables, BMI, ODI, EQ-5D-5L and HADS scores were comparable among groups (Appendix A).

### 3.3. Compliance and Treatment Credibility

It should be noted that three patients received an incorrectly parameterized VAGUSTIM device that was set to 20 instead of 30 min stimulation sessions. Two of them contacted the department less than 48 h after enrollment in the study, which corrected the problem, while one of these patients did not contact the department and the setting error was not discovered until the V2 visit. This patient had poor compliance and had only completed 13 stimulation sessions in the first 30 days. One patient was provided with an inoperable device with a faulty battery. It was not possible to replace it and the patient continued daily stimulation with his personal ECO 2 TENS stimulator, which was set in the same way as the VAGUSTIM (25 Hz, 30 min per day). However, it was not possible to check the patient’s compliance by interrogating his personal device, and we were only able to record the nine stimulations performed with the defective VAGUSTIM.

Although the 30 min duration per session was considered acceptable by the patients, daily compliance to treatment was not the same for all patients. After compiling the stimulation history of each device, the average compliance was 23.2 stimulations (out of a possible 30) after 1 month and 58.8 stimulations (out of a possible 90) after 3 months. A total of 16 out of 27 patients (59%) reached the 80% threshold after 1 month, and 13 out of 24 patients (54%) reached it after 3 months. Therefore, we compared the magnitude of change in VAS for low back pain in the subgroups with and without compliance. After 1 month, the VAS change was −16.9 (SD = 17.9) in the non-compliant group and −15.5 (18.4) in the compliant group (*p* = 0.84). After 3 months, the VAS change was −18.7 (28.9) in the treatment non-compliant group and −25.7 (21.7) in the treatment compliant group, although the difference was not significant (*p* = 0.51).

Treatment credibility ratings were overall satisfactory, with a mean treatment credibility rating of 70.4 (13%), although we did not find a significant change in this parameter between the 1- and 3-month use of the device.

### 3.4. Heart Rate Variability (HRV) Parameters

We observed a decrease in the mean RR interval (RRI) and the standard statistical measure of HRV (Root Mean Square of Successive Differences of RRI) after three months of taVNS. We did not detect any other statistically significant changes, although there was a trend towards an increase in high frequency power at 3 months, but this may have been masked by an excessively large standard deviation. The data are presented in Appendix A.

### 3.5. Tolerance

The treatment was well tolerated, and no serious adverse events occurred. A total of six adverse events were recorded, of which four were of minor severity and two of mild severity; two of the six adverse events could be attributed to the use of taVNS (Table 3). However, the following incidents, which have already been mentioned, should be discussed: One patient experienced exacerbation of psoriasis on the pinna, although taVNS has not been associated with any type of pro-inflammatory event in the literature [28]. A second patient had multiple hypertensive episodes for which there were other possible explanations. It is not known that taVNS causes hypertensive episodes, but that it lowers systolic and diastolic blood pressure [29]. Nevertheless, this patient was excluded from the analysis after the 1-month visit.

## 4. Discussion

In this proof-of-concept pilot study in order to assess the feasibility of taVNS in CLBP, we observed a statistically significant improvement in the pain VAS after 1 and 3 months of stimulation for this small sample of patients. We also observed a significant improvement in the degree of disability, quality of life and catastrophizing as assessed by patient-reported outcomes. The tolerability of the treatment was good, with no serious adverse events; six patients reported minor-to-mild adverse events, some of which were most likely not related to taVNS. One of the strengths of the study was that use of the device could be tracked throughout the follow-up period. The compliance rate was acceptable, with 54% of the patients having a compliance rate of more than 80% over the 3-month period and a mean compliance rate of 66%. These values are concordant with other studies exploring taVNS effects in other conditions, even in the long term [30,31], but can be qualified by the fact that patients received a weekly telephone call during the first month. Comprehensive information and expectation management, as well as regular monitoring through digital reminders could help address this challenge [32]. Nevertheless, this may have undermined the effect of taVNS on pain symptoms, although we found no statistically significant difference between patients who complied with the device and those who did not. Accordingly, taVNS appears to be suitable for long-term use in a clinical setting, whereas chronic pain patients often have problems with adherence to treatment protocols, especially when therapies are burdensome or poorly tolerated [33].

Our findings are consistent with previous research showing the efficacy of taVNS in other chronic pain conditions, such as fibromyalgia and migraine [14]. Several studies have reported improvements in pain scores and quality of life following taVNS therapy for these conditions [15]. While the exact mechanisms are still under investigation, these studies suggest that taVNS may modulate pain pathways and reduce neuroinflammation, which may also be true for CLBP [16]. The ability of the vagus nerve to activate parasympathetic pathways may counteract the sympathetic overactivity that often occurs in chronic pain conditions, thus restoring autonomic balance [29,34]. This shift may lead to reduced neuroinflammation, which is known to contribute to chronic pain syndromes [35]. In addition, functional neuroimaging studies have shown that taVNS stimulates the nucleus tractus solitarius (NTS), the first central relay of the vagus nerve [36]. The NTS is connected to important pain-processing regions in the brain, including the thalamus and limbic structures involved in the perception and emotional response to pain [8]. By modulating these circuits, taVNS can reduce both the sensory and affective components of pain [37]. The significant improvements in pain intensity observed in our patients are consistent with this neurophysiological model, suggesting that taVNS has the potential to influence multiple aspects of the pain experience. Compared to other therapies for CLBP, taVNS offers several advantages as it is a non-invasive, easily administered technique that can be well tolerated by patients, even with minimal clinical monitoring. It could be a particularly attractive option for patients with chronic pain who may have concerns about the risks and side effects of more invasive procedures [5]. In addition, the low number of adverse events in our case series contrasts with pharmacologic treatments such as opioids or nonsteroidal anti-inflammatory drugs, which carry risks of dependence, gastrointestinal problems and cardiovascular side effects, particularly with long-term use [4,38,39].

However, the design of the study does not allow us to confirm the efficacy of taVNS in CLBP, as we cannot exclude a contextual effect that could explain a large part of the clinical course of this patient group. Indeed, contextual effects play a crucial role in the treatment of chronic musculoskeletal (MSK) pain and significantly influence treatment outcomes beyond the specific intervention [40]. These effects include factors such as patient–therapist interaction, patient expectations and the therapeutic environment [41]. For example, studies show that immediate pain relief from treatments such as mobilization can be largely attributed to these non-specific factors [40]. The placebo effect, an integral part of contextual effects, illustrates how patients’ expectations can bring about real physiological changes, particularly in conditions such as CLBP, where psychological and emotional factors are closely linked to physical symptoms [42]. Positive communication from the therapist, such as affirmation of treatment success, can enhance pain relief, while neutral or negative statements can diminish this effect [43]. Here, the communication surrounding the information given to the patient could have induced positive expectations for the patients. In addition, the rituals of the treatment itself, whether through manual techniques or the use of therapeutic devices, can also contribute to the healing process by increasing the patient’s confidence [44].

Despite these promising results, our study has several limitations that should be considered. First, the lack of a control group means that we cannot definitively attribute the observed improvements to taVNS alone; a large randomized clinical trial is needed to confirm these results and assess the broader applicability of taVNS in CLBP populations. A sham taVNS control group will be necessary; among the several existing approaches in the literature, a strictly identical device to the one of the experimental group, with the same settings, but without electrical stimulation, seems to ensure satisfactory blinding conditions for the majority of patients [45]. This future clinical trial should also include further time points to assess the long-term and potential residual effects of taVNS. Another limitation is the reliance on patient-reported outcomes such as pain intensity and functional capacity, which may be influenced by subjective factors and may not fully capture the complexity of the pain experience. Objective measures, such as biomarkers of inflammation or clinical research on central or peripheral pain modulation, would provide valuable additional data to confirm our findings. Furthermore, we were unable to detect any significant changes in the HRV parameters. Although this biomarker could reflect the balance of the autonomic nervous system, its interpretation is complex and subject to various influences [46]. Environmental stress factors (e.g., hospital environment, physical activity) and individual variations in baseline autonomic function could lead to measurement noise. Although we tried to standardize the conditions by allowing participants to rest before measurement, external factors may still have influenced the reliability of HRV [25]. Future studies should consider more controlled environments for HRV data collection and use multiple measurement points (e.g., before and after each treatment session) to reduce variability and assess changes over time [47]. Other markers of autonomic function should also be investigated. Future studies should aim to incorporate such measurements to better elucidate the mechanisms underlying the therapeutic effects of taVNS. It would also be important to obtain a more comprehensive overview of the participants’ comorbidities, as these may influence both the treatment and evolution of CLBP [48]. Finally, patients were free to adjust the intensity of taVNS to achieve pain-free stimulation. Currently, there is no official recommendation for the minimum stimulation intensity [49], but it would be interesting to evaluate the efficacy of taVNS depending on different and more tightened stimulation intensities [29].

## 5. Conclusions

This pilot study suggests that 3 months of taVNS in patients with chronic low back pain is feasible, safe and could lead to the improvement of pain intensity and disability. A well-designed randomized controlled trial versus sham stimulation is needed to con-firm these encouraging results and strengthen the non-pharmacological range of treatments for this common pathology.

## Figures and Tables

**Figure 1 jcm-13-07601-f001:**
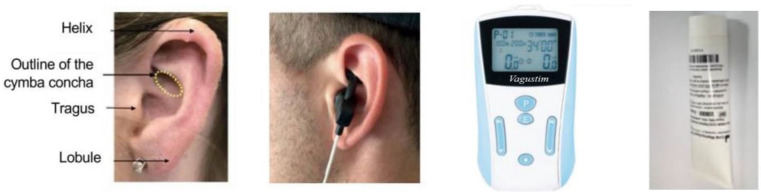
TaVNS device kit. Left to right: Description of the cymba concha, electrode positioning on the cymba concha, a TENSeco2 device (VAGUSTIM) and the conductive gel (copyright Schwa-Medico, adapted from Courties et al. (2022) [16]).

**Figure 2 jcm-13-07601-f002:**
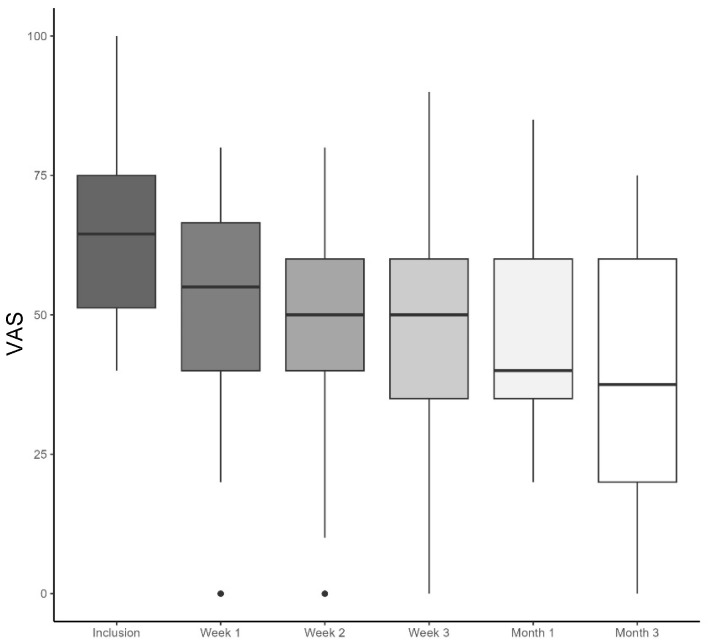
A box-and-whisker plot illustrating the evolution of low back pain VAS at any time point, including the weekly phone calls during the first month. The plot shows the median (bold line within the box), the interquartile range (box edges), the minimum and maximum values within 1.5 times the interquartile range (whiskers) and outliers (individual data points outside this range).

**Table 1 jcm-13-07601-t001:** The baseline characteristics of the patients included in the VALOM study.

	Baseline Characteristics (n = 30)
Age, years (SD)	47.8 (13.6)
Sex	
Men, n (%)	12 (40)
Women, n (%)	18 (60)
Work situation, n (%)	
In employment	18 (60)
Work stoppage	5 (16.7)
Unemployment	5 (16.7)
Retirement	6 (20)
Work disability	1 (3.3)
Duration of low back pain, months (SD)	141.3 (116)
Current analgesic consumption, n (%)	16 (53.3)
Opioid consumption, n (%)	15 (50)
History of lumbar steroid injection (epidural or facet joint), n (%)	16 (53.3)
History of spinal surgery, n (%)	6 (20)
>1 h per week of physical activity, n (%)	17 (56.7)
Body mass index, kg/m^2^ (SD)	26.2 (5.7)
Systolic blood pressure, mmHg (SD)	124.9 (18.3)
Diastolic blood pressure, mmHg (SD)	70.5 (10.3)
Clinical scores	
Low back pain VAS, mean (SD)	64 (13.9)
ODI, mean (SD)	37 (15.7)
PCS total score, mean (SD)	28.3 (12.3)
EQ-5D-5L 10-cm VAS, mean (SD)	51.8 (19.4)
HADS total score (/42), mean (SD)	19 (6.3)
HADS anxiety subscore (/21), mean (SD)	11.3 (3.5)
HADS depression subscore (/21), mean (SD)	7.7 (3.7)

HADS: Hospital Anxiety and Depression Scale, ODI: Oswestry Disability Index, PCS: Pain Catastrophizing Scale.

**Table 2 jcm-13-07601-t002:** Efficacy of taVNS at 1 and 3 months.

**At 1 Month**						
	**N**	**Baseline**	**Follow-Up**	**Mean Change**	***p*-Value**	**Effect Size**
Low back pain VAS	27	63 (13.9)	46.9 (18)	−16.1 (17.9)	<0.001	0.9
ODI	27	33.8 (12.4)	26.9 (12.7)	−6.9 (9)	<0.001	0.77
EQ-5D-5L VAS	26	53.9 (17.2)	63 (18.5)	9.1 (21.5)	0.04	0.42
HADS total score	27	18.1 (6)	17 (4.8)	−1.1 (3.8)	0.14	0.29
PCS total score	22	25.3 (11.8)	19.7 (12.1)	−5.6 (8.6)	0.01	0.65
**At 3 Months**						
Low back pain VAS	24	61.1 (12.2)	38.6 (23.6)	−22.5 (25)	<0.001	0.90
ODI	24	33.2 (10.6)	21.3 (12.9)	−11.9 (11.1)	<0.001	1.07
EQ-5D-5L VAS	24	52.1 (16.6)	66.3 (16.1)	14.2 (22.4)	0.01	0.63
HADS total score	23	18.5 (6.2)	15.8 (6.3)	−2.7 (5)	0.02	0.54
PCS total score	21	27.1 (11.6)	18.5 (13.5)	−8.6 (9.9)	<0.001	0.87

Results are presented as mean (SD). Cohen’s *d* effect sizes for each outcome are also calculated.

**Table 3 jcm-13-07601-t003:** Reported adverse events over 3 months of taVNS in CLBP patients.

Severity	Adverse Event	Number of Patients	Related to Device (Yes/No/Uncertain)
Minor	Epistaxis	1	Uncertain
	Neck pain	1	Uncertain
	Palpitations, hypertension, unease	1	Yes
Mild	Psoriasis flare-up	1	Uncertain
	Fall with radius fracture	1	No

## Data Availability

The data may be obtained upon request to the corresponding author.

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
