# Peer review of "Effect of Transcutaneous Auricular Vagus Nerve Stimulation in Chronic Low Back Pain: A Pilot Study"

_jcm, 2024, doi:10.3390/jcm13247601_

Round 1

Reviewer 1 Report

Comments and Suggestions for Authors

Overview

I found this trial extremely well done. It was well written, the introduction was thorough, the statistics were appropriate, tables were good, the discussion was good, and the conclusions were appropriate. I only have a few minor comments. I congratulate the authors and feel if they make these changes, it will be a great publication.

Major comments

1.      None

Minor comments

1.      Could you please include an image of transcutaneous auricular vagus nerve stimulation, highlighting or describing any important methodological parts, equipment, or anatomy as it is applied to the participant? This can be a mock / demonstration figure, for de-identification purposes.

2.      You mention a follow-up trial using a sham in the conclusion. You could add a couple sentences in the Discussion about what this sham might involve. What are the different options for a sham and how would this be done? If this is not known, that is OK, but maybe give some suggestions (would the electricity be off, would it be in a different spot on the ear, etc.). I also wonder if you could do the intervention plus usual care versus usual care alone (VNS+UC versus UC) or maybe an NSAID (VNS+NSAID versus NSAID), etc. Sometimes it’s tough to do a sham so those options would allow for a design without a sham. Also, your study did not really validate the feasibility of using a sham to my understanding (unless this has been validated previously). Using your experience, this edit would be helpful to other researchers, or maybe even could be referenced in your follow-up trial on this topic.

3.      Supplemental Figure – it might be valuable to include this box and whisker plot in the main manuscript. I think it’s a really strong visual and the caption is also good. It helps explain the findings. The only addition would be to explain it as a box plot and describe the different markers in the caption like the median, quartiles, min/max, outliers, etc.

4.      Adverse events – I am unsure why “Earpiece failure” is listed as an adverse event. Did this cause the patient harm in any way? If it was simply replaced by a properly functioning device, I don’t see how this causes the patient any problem.

Comments for clarity

1.      Standard deviation – when writing out standard deviation in paragraphs, can you put (SD) or standard deviation before the first value? No need to do this for every instance, but maybe the first time? This will distinguish the values from confidence intervals or standard error. For example, the abstract could be “VAS scores decreased significantly by 16.1 (standard deviation = 17.9) mm”

2.      P-values. p values smaller than 0.001 can be reported as p<0.001. I think you are using scientific notation but it can be hard to read. See: Aguinis, Herman, Matt Vassar, and Cole Wayant. "On reporting and interpreting statistical significance and p values in medical research." BMJ Evidence-Based Medicine 26.2 (2021): 39-42.

3.      VALOM – can you explain what this acronym means at it’s first use? (or maybe it’s not an acronym?

4.      Minor typo here: “Analyzes on independent samples (observer vs. non-observer and responder vs. non-responder) were performed” – should be “Analyses”. Same comment for “Statistical analyzes were performed”

5.      Maybe a translation issue here – “The third patient suffered from an episode of debilitating sciatica, did not attend the one-month visit, and underwent recalibration surgery in the lumbar region” – should the term “recalibration” be “decompression” or “fusion” or a more standard surgical term?

6.      Maybe a translation issue again – “History of lumbar spine infiltration” or “infiltration” appearing elsewhere – should this be “epidural steroid injection”?

Author Response

Dear Reviewer,

We sincerely thank you for your thoughtful and constructive feedback on our manuscript. Your comments and suggestions have been invaluable in helping us refine and improve the manuscript.

In this response document, we address each of your comments in detail. For clarity, we have provided your original comments in bold followed by our responses. Changes made to the manuscript in response to these comments have been outlined, and relevant sections/pages are indicated.

Overview

I found this trial extremely well done. It was well written, the introduction was thorough, the statistics were appropriate, tables were good, the discussion was good, and the conclusions were appropriate. I only have a few minor comments. I congratulate the authors and feel if they make these changes, it will be a great publication.

Major comments

  1. None

Minor comments

  1. Could you please include an image of transcutaneous auricular vagus nerve stimulation, highlighting or describing any important methodological parts, equipment, or anatomy as it is applied to the participant? This can be a mock / demonstration figure, for de-identification purposes.

Response:

Thank you for your suggestion to include an image of the transcutaneous auricular vagus nerve stimulation (taVNS) setup. We agree that such a figure would enhance the clarity of the methodology and provide readers with a better understanding of the procedure.

We have added a demonstration figure (Figure 1) to the manuscript. The figure illustrates the key anatomical landmarks, the position of the electrode, and the equipment used during stimulation.

This figure has been incorporated in the Methods section on page 3, line 112 and is referenced in the text as follows: “Figure 1 demonstrates the placement of the taVNS device, showing electrode positioning relative to the auricular anatomy.”
Figure 1 caption has been added: “Figure 1. TaVNS device kit. Left to right: Description of the cymba concha, electrode posi-tioning on the cymba concha, a TENSeco2 device (VAGUSTIM), and the conductive gel (copyright Schwa Medico, adapted from Courties et al. (2022) [16]).”

  1. You mention a follow-up trial using a sham in the conclusion. You could add a couple sentences in the Discussion about what this sham might involve. What are the different options for a sham and how would this be done? If this is not known, that is OK, but maybe give some suggestions (would the electricity be off, would it be in a different spot on the ear, etc.). I also wonder if you could do the intervention plus usual care versus usual care alone (VNS+UC versus UC) or maybe an NSAID (VNS+NSAID versus NSAID), etc. Sometimes it’s tough to do a sham so those options would allow for a design without a sham. Also, your study did not really validate the feasibility of using a sham to my understanding (unless this has been validated previously). Using your experience, this edit would be helpful to other researchers, or maybe even could be referenced in your follow-up trial on this topic.

Response:
Thank you for raising this important point regarding the inclusion of a sham in follow-up trials and alternative study designs. We agree that this is a crucial area to address both for methodological rigor and for guiding future high-quality research. We acknowledge that our study did not explicitly validate the feasibility of a sham condition, though prior research on taVNS has explored sham designs effectively.
Among the several existing approaches, a strictly identical device between groups, but without electrical stimulation in the control groups seems to be a good option. Subjective symptoms can occur with placebo stimulation, as shown in a randomized controlled study of taVNS in depression (Hein et al., 2013, 10.1007/s00702-012-0908-6), where 70% of patients in the placebo arm thought they were in the stimulation group.

This discussion has been added to page 9, line 363 of the revised manuscript.

“A sham taVNS control group will be necessary: among the several existing approaches in the literature, a strictly identical device to the one of the experimental group, with the same settings, but without electrical stimulation, seems to ensure satisfactory blinding conditions for the majority of patients [45]. This future clinical trial should also include […]”

  1. Supplemental Figure – it might be valuable to include this box and whisker plot in the main manuscript. I think it’s a really strong visual and the caption is also good. It helps explain the findings. The only addition would be to explain it as a box plot and describe the different markers in the caption like the median, quartiles, min/max, outliers, etc.

Response:
Thank you for your positive feedback on the supplemental figure and for suggesting its inclusion in the main manuscript. We agree that it is a strong visual that effectively conveys the findings and enhances the clarity of the results.

We have moved the box-and-whisker plot from the Supplemental Materials to the main manuscript as Figure 2. Additionally, we revised the figure caption to include a brief explanation of the box plot elements, as follows:

“Figure 2. Box-and-whisker plot illustrating evolution of low back pain VAS at any timepoint. The plot shows the median (bold line within the box), the interquartile range (box edges), the minimum and maximum values within 1.5 times the interquartile range (whiskers), and outliers (individual data points outside this range).”

Please let us know if further revisions or additional details would be helpful.

  1. Adverse events – I am unsure why “Earpiece failure” is listed as an adverse event. Did this cause the patient harm in any way? If it was simply replaced by a properly functioning device, I don’t see how this causes the patient any problem.

Response:

Thank you for pointing out this concern regarding the classification of “earpiece failure” as an adverse event. We agree that as the failure of the earpiece did not result in any harm or discomfort to the participant and was promptly addressed by replacing the device, it may not be qualified as an adverse event under standard definitions.

Upon review, we have re-evaluated the classification of “earpiece failure.” This event has been removed from the list of adverse events in the manuscript, as it did not result in harm or discomfort to the patient.

We appreciate your observation and believe this revision improves the accuracy of our reporting.

Comments for clarity

  1. Standard deviation – when writing out standard deviation in paragraphs, can you put (SD) or standard deviation before the first value? No need to do this for every instance, but maybe the first time? This will distinguish the values from confidence intervals or standard error. For example, the abstract could be “VAS scores decreased significantly by 16.1 (standard deviation = 17.9) mm”

Response:

Thank you for pointing this out. We agree that clearly distinguishing standard deviation (SD) from other statistical measures enhances clarity. We have revised the manuscript to define standard deviation as "(SD)" the first time it is mentioned in the abstract, and in paragraphs 3.1, 3.2, and 3.3, as appropriate.

For example in the abstract: “VAS scores decreased significantly by 16.1 (SD = 17.9) mm (p < 0.001)”

  1. P-values. p values smaller than 0.001 can be reported as p<0.001. I think you are using scientific notation but it can be hard to read. See: Aguinis, Herman, Matt Vassar, and Cole Wayant. "On reporting and interpreting statistical significance and p values in medical research." BMJ Evidence-Based Medicine26.2 (2021): 39-42.

Response:

Thank you for this suggestion and for referencing relevant guidance. We agree that using "p < 0.001" could be clearer and more reader-friendly than scientific notation for extremely small p-values.
Accordingly, we have revised all instances in the manuscript to report p-values smaller than 0.001 as “p < 0.001.”

  1. VALOM – can you explain what this acronym means at it’s first use? (or maybe it’s not an acronym?

Response:

Thank you for highlighting this. “VALOM” is the acronym of our study, standing for “VAgus nerve stimulation for LOw back pain Management”. We have clarified its full form when mentioned in the manuscript (page 2, line 84 : “The VALOM study (VAgus nerve stimulation for LOw back pain Management) was […]”)

  1. Minor typo here: “Analyzes on independent samples (observer vs. non-observer and responder vs. non-responder) were performed” – should be “Analyses”. Same comment for “Statistical analyzes were performed”

Response:

Thank you for catching this typo. We have corrected “analyzes / analyzed” to “analyses / analysed” in all instances within the manuscript to ensure proper usage.

  1. Maybe a translation issue here – “The third patient suffered from an episode of debilitating sciatica, did not attend the one-month visit, and underwent recalibration surgery in the lumbar region” – should the term “recalibration” be “decompression” or “fusion” or a more standard surgical term?

Response:

We appreciate your observation regarding the term “recalibration surgery.”. We agree that this is  a translation issue. We have revised the text to use the correct and standard surgical term, which in this case is “decompression surgery”.
The updated sentence now reads (page 5, line 193): “The third patient suffered from an episode of debilitating sciatica, did not attend the one-month visit, and underwent decompression surgery in the lumbar region.”

  1. Maybe a translation issue again – “History of lumbar spine infiltration” or “infiltration” appearing elsewhere – should this be “epidural steroid injection”?

Response:

Thank you for bringing this to our attention. We agree that “infiltration” may not be the most precise or standard term. We have revised the text to replace “lumbar spine infiltration” wherever appropriate by “lumbar steroid injection (epidural or facet joint)”, ensuring consistency and accuracy in the terminology.
In Table 1 : “History of lumbar steroid injection (epidural or facet joint), n (%)”
In the Results section, page 6 line 247 “…and less frequent lumbar steroid injections…”

We have strived to incorporate all feedback to the best of our ability and believe the revised manuscript has been strengthened as a result. Please do not hesitate to let us know if further clarifications or adjustments are needed.

Thank you again for your insightful and constructive comments.

Reviewer 2 Report

Comments and Suggestions for Authors

- Did you distinguish between specific and non-specific pain in the chronic low back pain patients included in the study?

- Even though this is a pilot study, please provide justification for the chosen sample size.

- Measuring Heart Rate Variability (HRV) during hospital visits might be influenced by various external factors such as transport, hospital environment, and stress, which could affect the reliability of the HRV data. Could there be errors in measurement due to this? Shouldn't HRV be measured before and after the intervention instead?

- The discussion lacks sufficient elaboration on HRV. Please add a discussion on this topic.

Author Response

Dear Reviewer,

We sincerely thank you for your thoughtful and constructive feedback on our manuscript. Your comments and suggestions have been invaluable in helping us refine and improve the manuscript.

In this response document, we address each of your comments in detail. For clarity, we have provided your original comments in bold followed by our responses. Changes made to the manuscript in response to these comments have been outlined, and relevant sections/pages are indicated.

- Did you distinguish between specific and non-specific pain in the chronic low back pain patients included in the study?

 Response:

Thank you for raising this point. We confirm that all patients included in our study had non-specific chronic low back pain (CLBP). This was explicitly stated in the Methods section through the exclusion criteria: “Exclusion criteria were […] low back pain caused by certain diseases such as ankylosing spondylitis, spondylodiscitis, or cancer.” To clarify this further, we have revised 2 sentences in the Methods section to emphasize that only patients with non-specific low back pain were recruited. The revised text now reads:

“Eligible participants were adults aged 18 to 70 years who had suffered from non-specific CLBP” (page 2, line 86)”

“Exclusion criteria were […] and low back pain caused by specific diseases (such as ankylosing spondylitis, spondylodiscitis or cancer).” (page 2, line 96)

This ensures that our recruitment strategy is clearly communicated to readers.

- Even though this is a pilot study, please provide justification for the chosen sample size.

Response:
Thank you for pointing this out. We acknowledge that a clear justification for the sample size strengthens the study's rigor. As this is a pilot study, the primary aim was to assess feasibility and gather preliminary data to inform the design of a larger randomized controlled trial (RCT). The sample size of 30 participants was chosen based on practical considerations, including resource availability and recruitment feasibility. Moreover, we based this choice on recommendations in the literature, which suggest sample sizes of 20 to 30 participants are sufficient to assess feasibility and estimate variability, enabling the detection of medium-to-large effect sizes for planning larger trials (see Whitehead et al., 2015 10.1177/0962280215588241).

We have now elaborated on this rationale in the Methods section on page 4, line 171: ” A sample size of 30 participants was chosen based on practical considerations, and recommendations suggesting that 20 to 30 participants are sufficient to assess feasibility, estimate variability, and detect medium-to-large effect sizes in pilot studies (Whitehead et al., 2015).”

- Measuring Heart Rate Variability (HRV) during hospital visits might be influenced by various external factors such as transport, hospital environment, and stress, which could affect the reliability of the HRV data. Could there be errors in measurement due to this? Shouldn’t HRV be measured before and after the intervention instead?

Response:

Thank you for highlighting the potential influence of external factors on HRV measurement during hospital visits. We agree that environmental and situational factors, such as the ones you cited, could introduce variability in HRV data. While we attempted to minimize these effects by allowing participants to rest for a standardized duration before HRV measurements, we acknowledge that this may not completely eliminate these influences.

Your suggestion to measure HRV before and after each treatment session is well-taken. For this pilot study, we opted for a single-point measurement to assess feasibility, but we recognize that measuring HRV this way could provide a more robust and reliable assessment.

We have added this limitation to the Discussion section on page 9, line 372 as follows:

“Furthermore, we were unable to detect any significant changes in the HRV parameters. Although this biomarker could reflect the balance of the autonomic nervous system, its interpretation is complex and subject to various influences [46]. Environmental stress factors (e.g. hospital environment, physical activity) and individual variations in base-line autonomic function could lead to measurement noise.”

- The discussion lacks sufficient elaboration on HRV. Please add a discussion on this topic.

Response:

Thank you for this suggestion. We agree that a more detailed discussion of heart rate variability (HRV) is important to contextualize our findings. We have expanded the Discussion section in line with your two above comments, the revised text now reads on page 9, line 372:

“Furthermore, we were unable to detect any significant changes in the HRV parameters. Although this biomarker could reflect the balance of the autonomic nervous system, its interpretation is complex and subject to various influences [46]. Environmental stress factors (e.g. hospital environment, physical activity) and individual variations in baseline autonomic function could lead to measurement noise. Although we tried to standardise the conditions by allowing participants to rest before measurement, external factors may still have influenced the reliability of HRV [25]. Future studies should consider more controlled environments for HRV data collection and use multiple measurement points (e.g. before and after each treatment session) to reduce variability and assess changes over time [47].”

We have strived to incorporate all feedback to the best of our ability and believe the revised manuscript has been strengthened as a result. Please do not hesitate to let us know if further clarifications or adjustments are needed.

Thank you again for your insightful and constructive comments.
